# Recombinant Oil-Body-Expressed Oleosin-hFGF5 in *Arabidopsis thaliana* Regulates Hair Growth

**DOI:** 10.3390/genes14010021

**Published:** 2022-12-22

**Authors:** Hongyu Wang, Xinxin Lan, Muhammad Noman, Ze Wang, Jing Zhang

**Affiliations:** 1College of Life Sciences, Jilin Agricultural University, Changchun 130118, China; 2Key Laboratory of Straw Biology and Utilization, The Ministry of Education, Changchun 130118, China

**Keywords:** *Arabidopsis*, recombinant oil-body, oleosin-hFGF5, hair growth, RNA-Seq

## Abstract

FGF5 (Fibroblast Growth Factor) is a member of the fibroblast growth factor family, which not only regulates growth and development but also inhibits hair regeneration. The oil-body expression vector pOTB-hFGF5 was constructed by the genetic engineering method and it was transformed into *Arabidopsis* by flora dip. T3 homozygous transgenic *Arabidopsis* was obtained after screening and propagation by the PCR and Western blot methods. The recombinant oil-body-expressed oleosin-hFGF5 can inhibit the proliferation of hair follicle epithelial cells and it exhibits the pharmacological activity of inhibiting hair regeneration in vivo by protein hybridization and imunohistochemistry. At the same time, the potential mechanism of recombinant oil-body-expressed oleosin-hFGF5 inhibiting hair growth was also revealed by RNA-Seq. This implies that the recombinant oil-body-expressed oleosin-hFGF5 has a good effect on inhibiting hair growth.

## 1. Introduction

The oil body is a sub-organelle of 0.5–2.5 μm in oil crop seed [1], which is a spheroid composed of oil-body-associated proteins, triacylglycerol (TAG) and a monolayer of phospholipids [2]. Oil-body-associated proteins include caleosin, oleosin and steroleosin. Among them, oleosin is composed of the C-terminus of the variable region, the N-terminus of the variable region and the middle hydrophobic region [3]. Therefore, the oil body expression system was established based on the structural characteristics of oleosin, which is an efficient expression system for foreign proteins. The target gene is fused downstream of the oleosin gene and the promoter drives the expression of the oleosin-target gene in the transgenic seeds [4]. Since the fusion gene is expressed on the surface of the oil body, it is embedded with the target protein and can be separated by centrifugation. Meanwhile, based on its structural characteristics, the oil body has a function similar to that of liposome and can carry functional substances as the delivery carrier. For example, a new transdermal drug delivery system (oil-body-linked oleosin-hEGF) was established, which accelerates wound healing [5], and the specific expression of oleosin-hFGF10 in safflower oil body can not only promote the transdermal absorption but also significantly improve the efficacy of hFGF10 [6]. aFGF, bFGF and hEGF were expressed in the oil body using oleosin fusion technology, which has the effect of reducing the wound size and promoting wound healing [7,8,9]. The growth factors with poor stability are easily hydrolyzed by protease in vivo and need to be stored at low temperature. However, seeds lose a lot of water after maturation, which can weaken the hydrolysis of oil-body-related proteins by enzymes. Therefore, the growth factor can be expressed in seeds through oil body fusion technology and can be stored in seeds for a long time.

Fibroblast growth factor 5 (FGF5) is a member of the fibroblast growth factor family, which has different biological effects when expressed in different tissues [10,11]. FGF5 promotes embryonic development in the early stage and the regeneration of endothelial cells and accelerates the repair of vascular injury [12]. The most common biological function of FGF5 is to regulate hair growth, so it is an important hair growth regulator [13,14], and excessive hair growth occurs when FGF5 is knocked out [15]. The growth cycle of hair is mainly divided into three stages, which are the growth stage, the degenerative stage and the resting stage [16]. All stages are affected by various growth factors, and FGF5 is involved in the regulation of the hair growth cycle [17]. The control of hair growth length by FGF5 may be related to the control of the hair growth cycle [18]. It was confirmed that FGF5 could promote the hair follicle from the growth phase to the rest phase as soon as possible by the culture of hair follicle cells, so that the growth of the hair follicle was inhibited. FGF5 can control the hair growth cycle, but it is unstable and easy to be degraded. Therefore, the production method of FGF5 can be changed by oil body fusion technology to improve its stability.

In this study, the oleosin-hFGF5 fusion gene was expressed on the surface of the oil body in *Arabidopsis thaliana*. The activity of the recombinant oil body expressing hFGF5 was confirmed by cell proliferation in vitro and hair inhibition in vivo. At the same time, the mechanism of recombinant oil body expressing hFGF5 and inhibiting hair growth was analyzed to provide a reference for its application and development by RNA-Seq.

## 2. Materials and Methods

### 2.1. Construction of pOTB-rhFGF5 Vector

The human FGF5 (hFGF5) gene was optimized and synthesized by Sangon Biological Company (Gene ID: 2250). The gene of hFGF5 was digested by Nco I and Hind III and ligated to the pOTB vector using ligase T_4_, and the recombinant vector pOTB-hFGF5 was successfully constructed (Appendix A). It was transformed into Agrobacterium tumefaciens *EHA105* cells and the positive monoclonal strains were screened and identified by PCR to prepare engineering strains for infection. PCR-specific primers are as follows: forward primer 5′ CATATGCACGGGGAGA3′ and reserve primer 5′ AAGCTTATCCAAAGCG 3′. PCR reaction process: pre-denaturation at 95 °C for 5 min, denaturation at 95 °C for 30 s, anneal at 56 °C for 30 s, amplification for 30 cycles and extension at 72 °C for 10 min.

### 2.2. Arabidopsis Transformation and Screening of Transgenic Lines

*EHA105* cells were collected at 5000 rpm for 10 min and dissolved in floral dip solution with OD600 value up to 0.8. *Arabidopsis* was transformed by floral dip, and the inflorescences were soaked in the infection solution for 5 min (1 L infection solution containing 3.1 g MS powder, 50 g sucrose, 0.5 g MES powder, 0.05% silwet, 10 μg 6-BA) and cultured in dark conditions for 24 h, and then T1 seeds were harvested under normal light conditions. T1 seeds were sown to get 4–6 leaves, which were sprayed with 0.1% glyphosate solution and identified by PCR to screen the transgenic line. T3 transgenic seeds were harvested by propagation. A total of 0.05 g of transgenic seeds was grounded by PBS buffer, and it was centrifuged at 12,000 rpm for 5 min to collect the upper layer. The oil body was boiled and the protein expression was identified by Coomassie brilliant blue staining and Western blot. 

### 2.3. Recombinant Oil Body Extraction

The transgenic and wild-type seeds (each 200 mg) were placed in 1 mL of PBS buffer and ground for 3 min. The extraction procedures of the oil body were referred to the methods of Qiang et al. [9].

### 2.4. Cell Proliferation Experiment

The mouse vibrissa follicles were obtained from C57BL/6 suckling mice under aseptic conditions. The hair follicle tissue was digested by 0.1% type I collagenase for 15 min. Subsequently, the tissue was digested with 0.25% trypsin for 15 min, and the digestion was terminated by adding an appropriate amount of high-sugar DMEM medium. These cells were inoculated into the culture dishes pretreated with type I collagen. Cells were cultured for 6 passages and then treated with drugs, including the control group (CG), wild-type oil body group (WT), hFGF5 group (PG) and recombinant oil body expressing hFGF5 group (ROBF5). The cells were starved in a K-SFM culture medium containing 0.1% fetal bovine serum for 24 h. Additionally, then, the control group was treated with 0.01 M PBS, while the concentrations of WT, PG and ROBF5 were 0, 6.25, 12.5, 25, 50 and 100 ng/mL. The proliferation efficiency of cells was measured by MTT assay. MTT solution was added and incubated for 4 h at 37 °C, followed by adding 100 μL DMSO solution and incubation at room temperature for 10 min.

### 2.5. Establishment of a Hair Loss Mice Model and Drug Treatment

The male mice of C57BL/6 aged 6 weeks (20 g–22 g) were randomly divided into 5 groups (*n* = 6 in each group): control group (CG) with 200 μL 0.01 M PBS, wild-type oil body group (WT) with 200 μL wild-type oil body, recombinant oil-body-expressed hFGF5 group (ROBF5) with 3.1 mg transgenic oil body containing 5 μg hFGF5, hFGF5 positive group (PG) with 200 μL solution containing 5 μg hFGF5 and hair retardant cream group (HRC). The experimental animals were handled according to the “license for the use of laboratory animals” of Jilin Agricultural University (SYJK 2018-0023). Mice were anesthetized with isoflurane, and the back hair was shaved to form a 2 cm × 3 cm area to make a hair loss model. The frequency of administration was once a day. All animal experiments were approved by the Laboratory Animal Welfare and Ethics Committee of Jilin Agricultural University (No.20201204001).

### 2.6. H&E Staining Analysis

The skin of mice was fixed with 4% paraformaldehyde overnight, and embedded in paraffin after dehydration, transparency and embedding treatments for pathological analysis. Paraffin blocks were cut into 5 μm sections, dewaxed, hydrated and stained with an H&E staining kit. The shape and number of hair follicles and hair regeneration were observed under an optical microscope.

### 2.7. Immunohistochemical Staining

The sections were soaked into the sodium citrate solution of 0.01 M and heated and pressured for 5 min to repair the tissue antigens. The solution containing 80% methanol and 3% H_2_O_2_ was added to the sections for 15 min to inactivate endogenous horseradish peroxidase (HRP). The sections were treated with 1% Triton of PBS, and then the sections were blocked for 1 h by 5% BSA at room temperature and incubated overnight at 4 °C with polyclonal rabbit anti-cytokeratin 14 (Bioss, Shanghai, China, bsm-52054R, 1:200), and the goat anti-rabbit HRP was incubated for 1 h. The positive protein was stained by DAB kit, and coloration was observed and photographed under optical microscope.

### 2.8. Western Blot Experiment

The target protein was extracted and quantified by the BCA method. The protein was separated by 12% SDS polyacrylamide gel (SDS-PAGE), transferred to PVDF membrane and blocked for 120 min by 5% skimmed milk powder. The first antibodies were incubated overnight at 4 °C, washed 3 times, followed by incubation with the second antibody goat anti-rabbit for 120 min and washed 3 times. The target bands were visualized using electrochemiluminescence. The first antibodies were as follows: the rabbit anti FGF5 antibody (Bioss, Shanghai, China, bs-1257R, 1:1000), rabbit polyclonal anti-β-actin (Bioss, Shanghai, China, bs-0061R, 1:5000) and rabbit polyclonal cytokeratin 14 (Bioss, Shanghai, China, bsm-52054, 1:1000).

### 2.9. RNA-Seq Analysis

The skin from mice treated with the positive drug hFGF5 and recombinant oil body samples was collected and sent to Orvison gene, Beijing, China, for transcriptome sequencing analysis, which was used to explore the genes of recombinant oil body regulating hair growth and screen the related pathways. The expression of differential genes was analyzed and further verified by qRT-PCR method. The pathway of recombinant oil body expressing hFGF5 and regulating hair regeneration was explored to elucidate its mechanism.

### 2.10. Statistical Analysis

All experiments were replicated three times. The results were expressed as mean ± standard deviation. The experimental data were analyzed through Graph Pad Prism 6.01 software and statistics were performed by ANOVA which was conducted by one-way method, * *p* < 0.05, ** *p* < 0.01 and *** *p* < 0.001.

## 3. Results

### 3.1. Oleosin-hFGF5 Was Successfully Expressed in Oil Body of Arabidopsis

The positive seeds were screened with 0.1% glyphosate [7] (Yuanye Biotechnology, Shanghai) and using positive seed cDNA as template. hFGF5-specific primers were used for PCR identification, and the target band of hFGF5 was amplified at 756 bp (Figure 1a), and then T3 transgenic seeds were obtained by propagation. The oil body was extracted from the transgenic seeds and the target band was detected at 47 kDa by SDS-PAGE and Western blot (Figure 1b,c). A T3 transgenic seed was obtained via the identification of the target gene and protein of transgenic *Arabidopsis*.

### 3.2. Recombinant Oil-Body-Expressed hFGF5 Inhibits the Proliferation of Hair Follicle Epithelial Cells In Vitro

When hair enters the growth stage, hair follicle epithelial cells proliferate and differentiate rapidly under the regulation of various growth factors to form hair fibers, which is the basis of hair regeneration. The recombinant oil-body-expressed hFGF5 had no significant effect on the epithelial cells of hair follicles at very low concentrations, and there was no significant difference between all treatment groups. However, when the concentration was higher than 12.5 ng/mL, the inhibitory effect on hair follicle epithelial cells in various treatment groups is significantly different, and the inhibitory effect of PG and ROBF5 increased with the increase in concentration and was significantly higher than that of the CG and WT groups (Figure 2). ROBF5 significantly inhibited cell proliferation in a dose-dependent manner with the increase in protein concentration.

### 3.3. Recombinant Oil-Body-Expressed Oleosin-hFGF5 Inhibits Hair Regeneration in Mice In Vivo

The hair on the back of the mice began to turn black on the 10th day in the CG group, WT group and ROBF5 group, and the black hair on the back of the mice continued to grow in the CG group and WT group, but black hair growth was not obvious in the ROBF5 group, and it was also observed in the HRC group, while no black hair was observed in the PG group (Figure 3a), indicating that the hair regeneration was inhibited after the treatment with recombinant oil body on the 15th day. A histopathology analysis of the inhibition of hair growth showed that there were more hair follicles in the subcutaneous tissue of the CG group and WT group, and the black dots existed in the hair follicles, indicating that there was a small amount of hair regeneration in the hair follicles by H&E staining on the 10th day. On the contrary, the number of hair follicles was decreased, lacking black dots in the hair follicles in the PG group, ROBF5 group and HRC group, indicating that the hair regeneration was weaker. On the 15th day, the new hair in the hair follicles of CG and WT groups was further elongated, and the volume of the hair follicles was increased compared to the 10th day; however, the hair growth was still inhibited and there was no obvious new hair in the hair follicles in the PG, ROBF5 and HRC groups (Figure 3b). This implies that recombinant oil body had the activity of inhibiting hair growth.

### 3.4. Effect of Recombinant Oil Body Expressed hFGF5 on Cytokeratin 14

The cytokeratin 14 is an important factor in process of hair regeneration. The positive staining rate of cytokeratin 14 in the CG and WT groups was significantly higher than in the PG, ROBF5 and HRC groups on the 10th day by immunohistochemistry (Figure 4a). The expression of cytokeratin 14 in the ROBF5 group was lower than that in the CG and WT groups, and slightly higher than that in the PG and HRC groups (Figure 4a). On the 15th day, there was no significant difference between the CG and WT groups, but the expression of cytokeratin 14 was higher than in the PG, ROBF5 and HRC groups. The expression of cytokeratin 14 in the PG, ROBF5 and HRC groups on the 15th day was higher than on the 10th day. The expression of cytokeratin 14 in hair follicle cells after different treatments was detected by Western blot, which was consistent with the results of immunohistochemical staining (Figure 4b,c). It has been proved that recombinant oil body can inhibit hair growth.

### 3.5. Differentially Expressed Genes by the Treatment of Recombinant Oil-Body-Expressed hFGF5

Compared with the CG group, 1412 genes were up-regulated and 1566 genes were down-regulated in the ROBF5 group (Figure 5a), and compared with the WT group, 2086 genes were up-regulated and 2038 genes were down-regulated in the ROBF5 group (Figure 5b), and compared with the PG group, the expression of 625 genes was up-regulated and 601 genes were down-regulated in the ROBF5 group (Figure 5c). Differences in the transcriptional level of the genes lead to changes in metabolic regulation after the treatment of the PG group and ROBF5 group, which ultimately produce different effects. Compared with the WT group, the up-regulated genes and the down-regulated genes in the ROBF5 treatment group were involved in the regulation of different signaling pathways by KEGG analysis (Appendix A).

### 3.6. The Mechanism of Recombinant Oil-Body-Expressed hFGF5 Inhibiting Hair Growth

Oil body treatment can affect the expression of key genes in the EGFR, TGF-β and Wnt signaling pathways, and then regulate hair growth. Compared with the WT treatment group, some differentially up-regulated genes in the ROBF5 group, such as E2F4, SMAD7, Bax, E2F5 and CSNK1E were involved in the regulation of the EGFR, TGF-β and Wnt pathways (Figure 6). Bax is mainly involved in the regulation of the EGFR pathway, which causes the activation of the intracellular apoptosis pathway and inhibits hair growth (Appendix A). The expression of Bax was up-regulated after ROBF5 treatment. E2F4, E2F5 and Smad7 are mainly involved in TGF-β, and Smad7 inhibits the TGF-β pathway by regulating the activation of Smad2/3. Additionally, the up-regulation of E2F4 and E2F5 directly inhibits the cell cycle, leading to cell apoptosis (Appendix A). CSNK1 is mainly involved in the regulation of the Wnt pathway, which inhibits Wnt/β-catenin signal transduction. The activation of the Wnt/β-catenin pathway was a classic intracellular signal transduction pathway regulating hair growth (Appendix A). Some differentially up-regulated genes such as ID2, SOS1 and PIK3R3 were involved in the regulation of the EGFR and TGF-β pathways (Figure 6). PIK3R3 was involved in the regulation of the EGFR pathway, which can promote the synthesis of intracellular proteins and cell proliferation (Appendix A). Compared with the WT group, the recombinant oil body down-regulated PIK3R3 in the EGFR pathway (Figure 6). SOS1 is involved in the regulation of the EGFR pathway, which promotes the genes expression of cell proliferation in the EGFR pathway, such as the ERK gene. The overexpression of SOS1 can promote the activation of ERK and enhance the activity of cell mitosis and differentiation (Appendix A). Compared with the WT group, the recombinant oil body down-regulated SOS1, which could inhibit hair growth (Figure 6). ID2 participated in the TGF-β pathway, which can promote DNA synthesis and accelerate cell proliferation and differentiation. Compared with the WT group, the recombinant oil body down-regulated ID2 in the TGF-β pathway. The intracellular signal transduction was changed after the treatment of the recombinant oil body, which affected the EGFR, TGF-β and Wnt pathways, and then inhibited hair growth. 

## 4. Discussion

The half-life of growth factor is too short to show effective biological activity in treatment, so it is particularly important to find effective material carriers [7]. Oil body as a new type of safe carrier material has been widely considered. The recombinant proteins expressed in the surface of oil body are easier to purify and have a longer half-life compared with the prokaryotic expression system. The oil body expression system is easy to store and transport, easy to process, reduces the cost of purification and most importantly, presents low toxicity, among others. Therefore, an oil-body expression system was selected to produce FGF5. Although hFGF5 is expressed in *A. thaliana*, its expression level is only about 0.19%. In the future, we need to further modify the gene and vector through genetic engineering technology to express in oil crops, so as to obtain transgenic seeds with high expression. The oleosin-rhFGF5 was expressed on the surface of the oil body, and the recombinant oil body expressing hFGF5 was proved to have the effect of inhibiting hair growth by cell and animal experiments. The growth of hair was inhibited, and the damage of hair follicles was also significantly attenuated, while the volume of hair follicles in subcutaneous tissue was not significantly reduced after treatment with recombinant oil body on the 15th day. This may be an advantage for the use of recombinant oil body expressing hFGF5 to temporarily inhibit hair growth on diseased skin without causing damage to hair follicles.

The differentially expressed genes screened by RNA-seq after recombinant oil body treatment were involved in the EGFR, TGF-β and Wnt pathways to regulate hair growth, which promoted cell proliferation and apoptosis. Among them, Bax was a classic gene regulating apoptosis, which is up-regulated expression that can cause apoptosis [19]. The overexpression of Smad7 can not only inhibit TGF-β signal transmission but can also affect the development process of hair follicles. When Smad7 in large quantities was expressed in hair follicle cells, the hair circulation was blocked from entering the growth period, leading to hair follicle degeneration [20]. The CSNK1E gene was one of the important candidate genes potentially regulating hair follicle development by sequencing the genome of sheep developing hair follicles [21]. ID2 can be highly expressed in the specialized cells in the basal epidermis and cytoplasm of the outer root sheath [22]. The hair growth-related pathway and immune-related pathway were abnormally activated by the transcriptome of NIH hairless mice, and the PIK3R3 gene was expressed in large quantities around the skin hair follicles of hairless mice [23]. PIK3R3 may be used as an important target for the potential regulation of hair growth. EGFR controls some genes that are involved in epidermal differentiation, cell cycle and apoptosis [24]. The loss of EGFR leads to absence of the LEF1 protein, specifically in the innermost epithelial hair layers. After activating the Wnt pathway, it binds to the receptor on the cell membrane and initiates a cascade reaction to up-regulate the expression of intracellular β-catenin, and the interaction between β-catenin and the N-terminus of the intracellular LEF/TCF transcription factor blocks the expression of downstream target genes, resulting in the hair going through a degeneration stage. TGF-β2 dominates the hair degeneration cascade, which prompts epithelial cells to up-regulate and activate stromal cell-secreted protein smad 9 and smad3, leading to the loss of epithelial cells and inducing hair to enter the degenerative stage. The recombinant oil body expressing hFGF5 could regulate the changes of these genes that play a role in hair growth.

## Figures and Tables

**Figure 1 genes-14-00021-f001:**
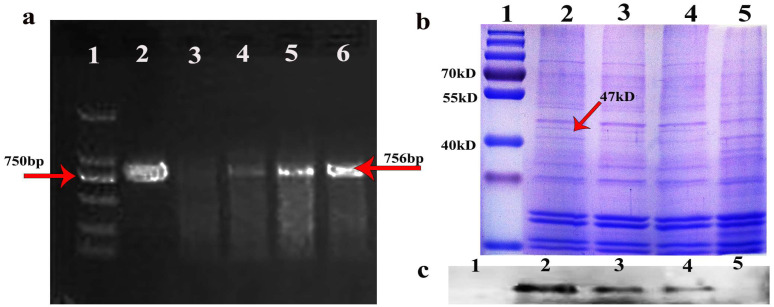
**Identification of transgenic *Arabidopsis*.** (**a**) PCR-identified using genomic DNA as a template and hFGF5-specific primers. line1: DNA marker, line2: hFGF5 positive control, line3: wild type *Arabidopsis* genomic as a negative control, line4–6: transgenic *Arabidopsis* genomic DNA as a template and hFGF5-specific primers (Red arrow (right) indicates the target band of 756 bp). (**b**) SDS-PAGE of the transgenic *Arabidopsis* (Red arrow indicates the expressed protein of 47 kD). (**c**) Western blotting identified the oleosin-hFGF5 fusion protein. line1: protein mark, line2–4: fusion protein from transgenic *Arabidopsis*, line5: protein from wild type *Arabidopsis*.

**Figure 2 genes-14-00021-f002:**
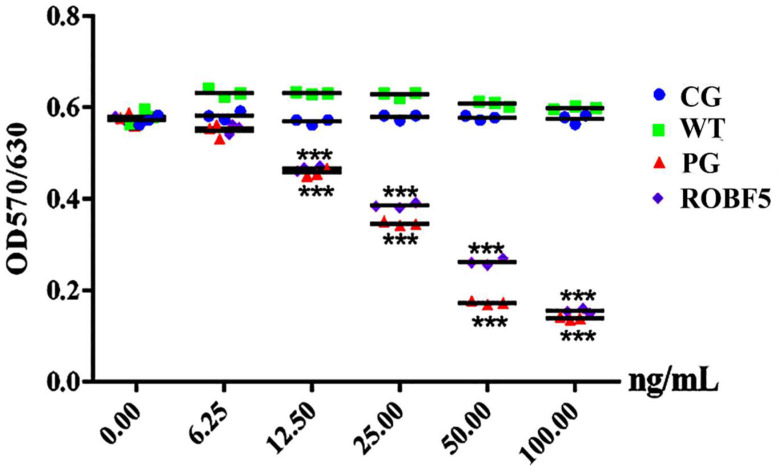
**Effects of recombinant oil body on the inhibition of hair follicle epithelial proliferation** (*** *p* < 0.001). CG: Control group (PBS), WT: wide-type oil body, PG: positive group (hFGF5), ROBF5: recombinant oil body expressing hFGF5.

**Figure 3 genes-14-00021-f003:**
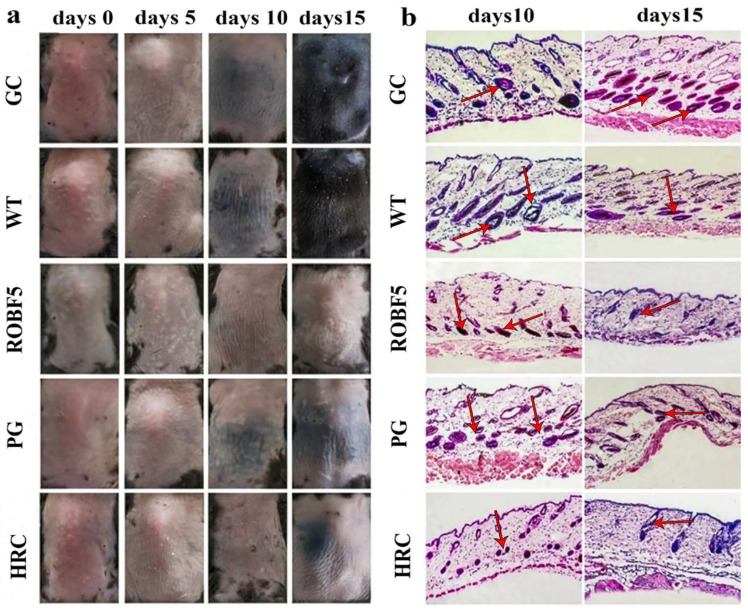
**Effect on inhibiting hair growth by recombinant oil body expressing hFGF5 in vivo.** (**a**) Hair regeneration on the back of mice in 15 days after administration (*n* = 6). (**b**) Pathological analysis by H&E staining on day 10 and day 15 (magnification 100 times). CG: Control group (PBS), WT: wide-type oil body, PG: positive group (hFGF5), ROBF5: recombinant oil body expressing hFGF5, HRC: hair retardant cream (Red arrows show the staining).

**Figure 4 genes-14-00021-f004:**
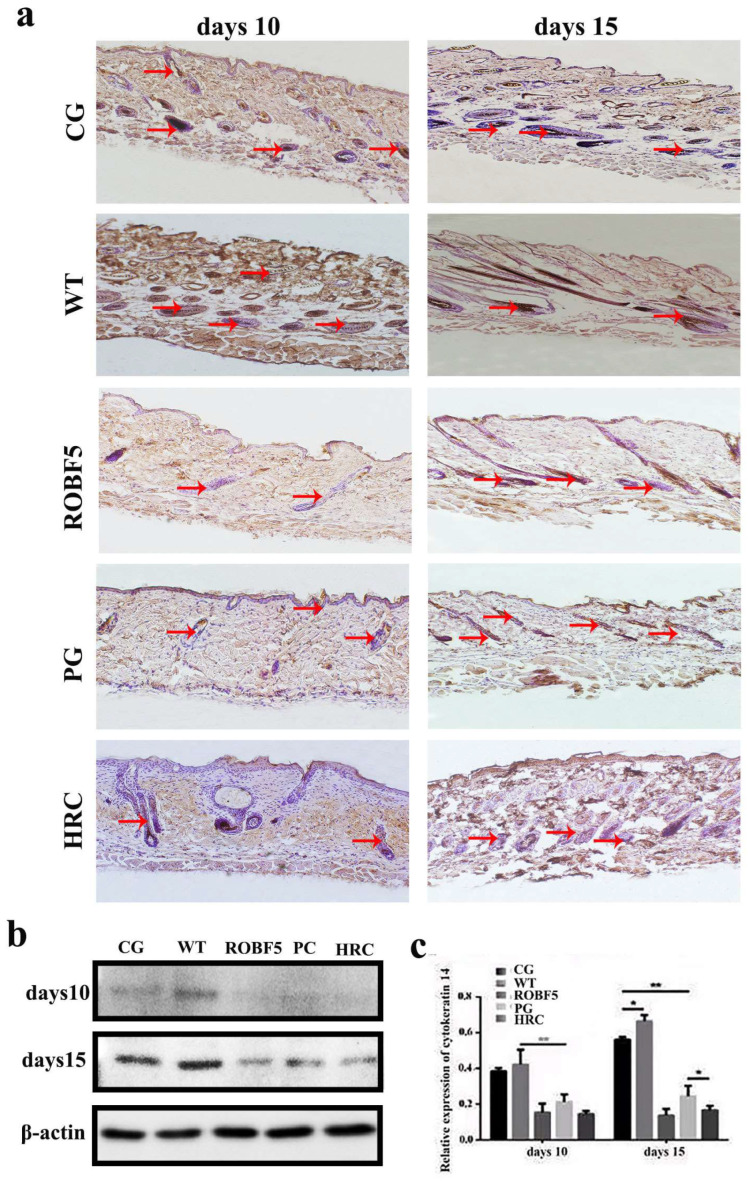
**The expression of cytokeratin 14 analyzed in vivo by immunohistochemistry staining and Western blotting.** (**a**) Immunohistochemistry staining of cytokeratin 14 (magnification of 100×), the red arrow indicates a positive protein. (**b**) Western blot analyzed the expression of cytokeratin 14. (**c**) The relative expression was calculated by the gray value of the Western blot (* *p* < 0.05, ** *p* < 0.01). CG: Control group (PBS), WT: wide-type oil body, PG: positive group (hFGF5), ROBF5: recombinant oil body expressing hFGF5, HRC: hair retardant cream.

**Figure 5 genes-14-00021-f005:**
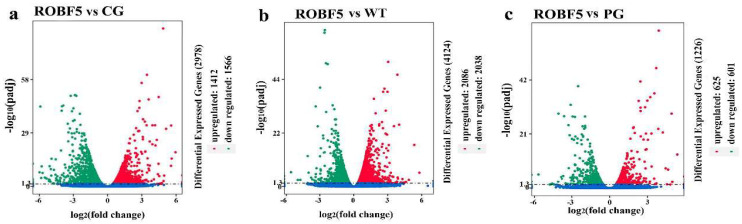
**Volcanic map of differential gene expression after treatment of recombinant oil body expressing hFGF5, compared with the treatment of CG, WT and PG.** (**a**) ROBF5 vs CG, (**b**) ROBG5 vs WT, (**c**) ROBF5 vs PG. CG: positive control group; WT: wild-type oil body treatment group; PG: hFGF5 treatment group; ROBF5: recombinant oil body expressing hFGF5 treatment group.

**Figure 6 genes-14-00021-f006:**
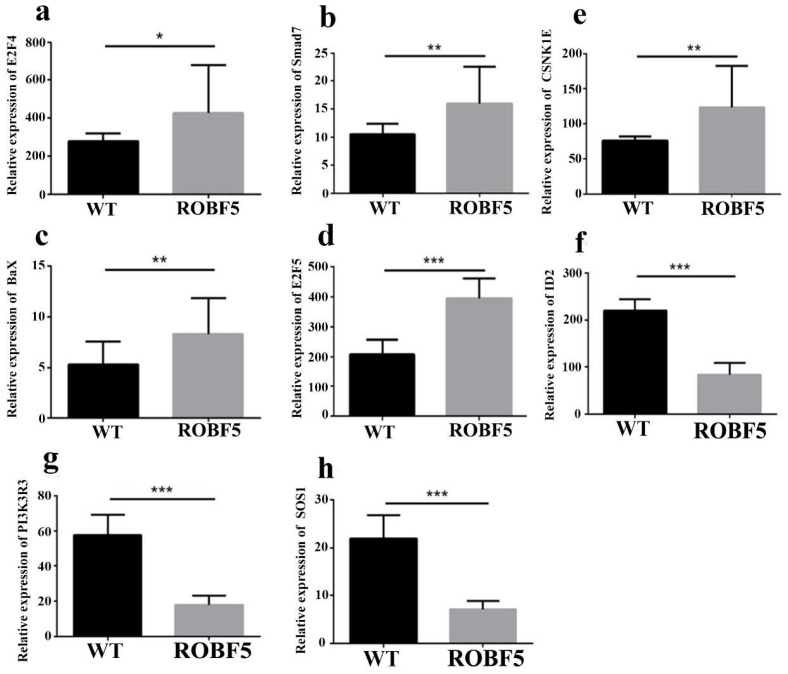
**The expression analysis of up-regulated and down-regulated differential genes after treatment of recombinant oil body.** (**a**) E2F4 gene relative expression, (**b**) Smad7 gene relative expression, (**c**) Bax gene relative expression, (**d**) E2F5 gene expression, (**e**) CSNK1E gene relative expression, (**f**) ID2 gene relative expression, (**g**) PI3K3R3 gene relative expression, (**h**) SOS1 gene relative expression (* *p* < 0.05, ** *p* < 0.01, *** *p* < 0.001). ROBF5: recombinant oil body expressing hFGF5 group, WT: wild-type oil body group.

## Data Availability

All data generated or analyzed during this study are available within.

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
