# Peer review of "Recombinant Oil-Body-Expressed Oleosin-hFGF5 in Arabidopsis thaliana Regulates Hair Growth"

_genes, 2022, doi:10.3390/genes14010021_

Round 1

Reviewer 1 Report

The oleosin has become an efficient expression system for foreign proteins. Genes of interest can be fused to the downstream of oleosin gene and be expressed in the oil body of transgenic seeds, which is developed to novel transdermal drug delivery system. The manuscript constructed hFGF5 expressing vector, extracted oil bodies from transgenic seeds, and tested the biological function on mice hair growth and development in vitro and in vivo. I found the manuscript interesting.

Major comments:

Figure 1. be sure to use the highest-quality figures possible. The way denoted target bands in Fig. 1a and Fig. 1b were misleading. Resolution of Fig. 1c were insufficient. Do the transgenic plants display morphological difference from wild-type plants, show images of transgenic plants. In the end, among the three transgenic lines selected, which one was used for recombinant oil body extraction. Do they have the same recombinant protein productivity?

Figure 2. Define the abbreviations. Give details in how statistical analysis was performed? Significant differences among which groups?

Figure 3. Denote air follicles in Fig.3b.

Figure 4. Change the color of indicative arrows in Fig. 4a. Labeling in Fig. 4b was misleading. The relative expression was calculated from how many western blot images as error bars were used in Fig. 4c. Provide original unprocessed images (and replicates) for peer-review process and attached the images as in supplementary.

Figure 5. Provide tables of differentially expressed genes; upload the RNA-seq data to public accessible repository, such as Gene Expression Omnibus (https://www.ncbi.nlm.nih.gov/geo/). Without the data, the quality and reliability of the results cannot be judged.

Figure 6. Primers for qRT-PCR were missing.

L65: provide details on the source of glyphosate and references that apply 0.1% glyphosate for screening of transgenic plants.

Ethics statement on the use of animals for research was missing.

Author Response

Reviewer 1

The oleosin has become an efficient expression system for foreign proteins. Genes of interest can be fused to the downstream of oleosin gene and be expressed in the oil body of transgenic seeds, which is developed to novel transdermal drug delivery system. The manuscript constructed hFGF5 expressing vector, extracted oil bodies from transgenic seeds, and tested the biological function on mice hair growth and development in vitro and in vivo. I found the manuscript interesting.

Major comments:

  1. Figure 1. be sure to use the highest-quality figures possible. The way denoted target bands in 1a and Fig. 1b were misleading. Resolution of Fig. 1c were insufficient. Do the transgenic plants display morphological difference from wild-type plants, show images of transgenic plants. In the end, among the three transgenic lines selected, which one was used for recombinant oil body extraction. Do they have the same recombinant protein productivity? 

Response: We appreciate your valuable suggestions and advices on our manuscript. It is very important and helpful to us, and I have accepted your suggestion and made the changes in the revised manuscript. The representation of the target frequency band in Fig. 1a and Fig. 1b has been modified, and the resolution of Fig. 1c has been improved. There was no significant morphological difference between the transgenic plants and the wild-type plants, so no morphological images were shown. Because they have different expression levels, the recombinant protein productivity is different. We selected the highest expression level from the three transgenic plants for the follow-up experiment to extract the recombinant oil body.

  1. Figure 2. Define the abbreviations. Give details in how statistical analysis was performed? Significant differences among which groups?

Response: The abbreviations were defined and supplemented in the revised manuscript. All experiments were replicated three times. The results were expressed as mean ± standard deviation. The experimental data were analyzed through GraphPad Prism 6.01 software and statistics was performed by ANOVA,which was conducted by one-way method. The recombinant oil body expressed hFGF5 had no significant effect on the epithelial cells of hair follicles at very low concentration, and there was no significant difference between all treatment groups. However, when the concentration was higher than 12.5 ng/mL, the inhibitory effect on hair follicle epithelial cells in various treatment groups is significantly different, and the inhibitory effect of PG and ROBF5 increased with the increase of concentration, and was significantly higher than that of the CG and WT groups (Fig. 2). ROBF5 significantly inhibited cell proliferation in a dose-dependent manner with the increase of protein concentration.

  1. Figure 3. Denote air follicles in Fig.3b.

Response: The air follicles were denoted in Fig.3b.

  1. Figure 4. Change the color of indicative arrows in Fig. 4a. Labeling in Fig. 4b was misleading. The relative expression was calculated from how many western blot images as error bars were used in Fig. 4c. Provide original unprocessed images (and replicates) for peer-review process and attached the images as in supplementary.

Response: In Fig.4a, the indicating arrow color has been changed to red. In Fig. 4b the images were revised. In Figure 4c, the relative expression was calculated by using the grayscale value of the target band, and the grayscale value of the reference protein was used as the standard. We repeated three times to calculate the gray value for the determination of relative expression.

The original picture of Figure 4b is as follows:

  1. Figure 5. Provide tables of differentially expressed genes; upload the RNA-seq data to public accessible repository, such as Gene Expression Omnibus (https://www.ncbi.nlm.nih.gov/geo/). Without the data, the quality and reliability of the results cannot be judged.

Response: The RNA-seq data were uploaded to the Sequence Read Archive database,

https://submit.ncbi.nlm.nih.gov/subs/sra/ and Sequence

Read Archive (SRA) submission: SUB12343422.

  1. Figure 6. Primers for qRT-PCR were missing.

Response 6: qRT-PCR primers are supplemented in supplementary file. (Tab.S1 Sequence of primers)

  1. L65: provide details on the source of glyphosate and references that apply 0.1% glyphosate for screening of transgenic plants.

Response 7: The source of glyphosate and a reference that apply 0.1% glyphosate for screening of transgenic plants was added.

As follow: The positive seeds were screened with 0.1% glyphosate [7] (Shanghai Yuanye Biotechnology Co., LTD).

  1. Ethics statement on the use of animals for research was missing.

Response 8: Ethics statement on the use of animals for research was added.

As follow: The experimental animals were handled according to the "license for the use of laboratory animals" of Jilin Agricultural University (SYJK 2018-0023).

Reviewer 2 Report

The manuscript designed an expression system of Fgf5 and evaluated the biological activity of this recombinant Fgf5. Data showed that the recombinant Fgf5 could inhibit hair regeneration.

Several concerns should be addressed to improve this manuscript.

 1. Why did the author choose oil body expression system to produce this recombinant human FGF5? What are the advantages of this expression system, compared with prokaryotic expression system, such as Escherichia coli? This should be discussed in the manuscript. 

2. In the Figure 1c and 4b, the data quality of this western blotting should be improved. The commercialized standard rhFGF5 should be included as a positive control in Figure 1c. 

3. Oil body expressing FGF5 inhibits hair growth only by EGFR, TGF-β, and Wnt pathways? Are there other pathways involved?

 4. Please add that the relationship between EGFR, TGF-β, and Wnt pathways is in discussion?

5. The RNA seq data, such as KEGG analysis, should be presented in the Figure 5.

Author Response

Reviewer 2

Comments and Suggestions for Authors

The manuscript designed an expression system of Fgf5 and evaluated the biological activity of this recombinant Fgf5. Data showed that the recombinant Fgf5 could inhibit hair regeneration. Several concerns should be addressed to improve this manuscript.

  1. Why did the author choose oil body expression system to produce this recombinant human FGF5? What are the advantages of this expression system, compared with prokaryotic expression system, such as Escherichia coli? This should be discussed in the manuscript. 

Response: We appreciate your valuable suggestions and advices on our manuscript. It is very important and helpful to us, and I have accepted your suggestion and made the changes in the revised manuscript. The half-life of growth factor is too short to show effective biological activity in treatment, so it is particularly important to find effective material carriers [7]. Oil body as a new type of safe carrier material has been widely considered. The recombinant proteins expressed in the surface of oil body are easier to purify and have a longer half-life compared with prokaryotic expression system. The oil body expression system is easy to store and transport, easy to process, reduce the cost of purification, and most importantly, low toxicity and so on, therefore oil body system is chosen to express hFGF5.

  1. In the Figure 1c and 4b, the data quality of this western blotting should be improved. The commercialized standard rhFGF5 should be included as a positive comntrol in Figure 1c. 

Response: We appreciate your valuable suggestions and advices on our manuscript. The quality of Fig.1c has been revised. hFGF5 is expressed in fusion with oleoesin protein, the size of the fusion gene is 1275bp, and the molecular weight of the fusion protein is about 47kDa, while the molecular weight of rFGF5 is about 27.7kDa, which is inconsistent. Therefore, commercialized standard rFGF5 is not used as a comparison, but only compared with wild-type Arabidopsis in Fig.1c.

  1. Oil body expressing FGF5 inhibits hair growth only by EGFR, TGF-β, and Wnt pathways? Are there other pathways involved?

Response: RNA-Seq was used to explore the pathways related to the possible regulation of hair growth after rhFGF5 and oleosin-rhFGF5 treatment. Firstly, KEGG pathway enrichment analysis was carried out according to the differences in gene expression, and many potential regulation pathways of hair growth were successfully enriched. However, it was found that some differential genes were involved in the regulation of EGFR, TGF-β and Wnt pathways in the enrichment pathways, at the same time, other pathways have also been involved, but the EGFR, TGF-β and Wnt pathway regulates hair growth. We demonstrated that oleosin-rhFGF5 can regulate the expression of some genes, and participate in the hair growth regulation pathways EGFR, TGF-β, and Wnt pathways, and ultimately inhibit hair growth.

  1. Please add that the relationship between EGFR, TGF-β, and Wnt pathways is in discussion?

Response: EGFR controls some genes that involved in epidermal differentiation, cell cycle and apoptosis [24]. Loss of EGFR leads to absence of LEF1 protein specifically in the innermost epithelial hair layers. After activating Wnt pathway, it binds to the receptor on the cell membrane and initiates a cascade reaction to up-regulate the expression of intracellular β-catenin, and the interaction between β-catenin and the N-terminus of the intracellular LEF/TCF transcription factor blocks the expression of downstream target genes, resulting in the hair going through a degeneration stage. TGF-β2 dominates the hair degeneration cascade, which prompts epithelial cells to upregulate and activate stromal cell secreted protein smad 9 and smad3, leading to the loss of epithelial cells and inducing hair to enter the degenerative stage.

  1. The RNA seq data, such as KEGG analysis, should be presented in the Figure 5.

Response: The KEGG enrichment results were shown in supplementary file (Figure1S)

Reviewer 3 Report

This is a very well written manuscript, but I want to know the practical implication of this study.

Author Response

Reviewer 3:

Comments and Suggestions for Authors

This is a very well written manuscript, but I want to know the practical implication of this study.

Response: We appreciate your valuable suggestions and advices on our manuscript. It is very important and helpful to us. The volume of hair follicles in subcutaneous tissue was not significantly reduced after treatment with recombinant oil body. It may be an advantage for the use of recombinant oil body expressing hFGF5 to temporarily inhibit hair growth on diseased skin without causing damage to hair follicles.

Round 2

Reviewer 1 Report

The authors have made a series of changes to improve the readability and legibility of their manuscript. The revision does not significantly change the manuscript conclusions (The recombinant oil body expressed oleosin-hFGF5 can inhibit the proliferation of hair follicle epithelial cells). I think the authors did a great job and the current version is suitable to publish in GENES. However, I noticed that authorship has been changed in the revision, reasons and agreement for the change should be provided before publication.

Reviewer 2 Report

The response letter addressed all my concern.